# Contextual Ambient Occlusion

Andrey Titov*
École de technologie
supérieure
Concordia University

Marta Kersten-Oertel†
Concordia University

Simon Drouin‡
École de technologie supérieure

## ABSTRACT

In this paper, we present a new volumetric ambient occlusion algorithm called Contextual Ambient Occlusion (CAO) that supports real-time clipping. The algorithm produces ambient occlusion images of exactly the same quality as Local Ambient Occlusion (LAO) while enabling real-time modification to the shape used to clip the volume. The main idea of the algorithm is that clipping only affects the ambient value of a small number of voxels, so by identifying these voxels and recalculating the ambient factor only for them, it is possible to significantly increase the rendering performance (by 2-5x) without decreasing the quality of the rendered image. Due to its fast performance, the algorithm is suitable for interactive environments where clipping changes could occur every frame. Additionally, the algorithm doesn't have any stereoscopic inconsistency, which makes it suitable for mixed reality environments.

**Index Terms:** Computing methodologies—Computer graphics—Rendering; Applied computing—Life and medical sciences—Computational biology—Imaging

## 1 INTRODUCTION

Ambient occlusion (AO) is a global illumination technique in computer graphics that is used to estimate how much each particle in the scene is illuminated with ambient lighting. The ambient factor of a particle can be viewed as an "accessibility" of the particle [9]. The general idea is that particles that are more obstructed and thus less accessible receive less ambient light and become darker consequently. In volume rendering, the particle is surrounded with semi-transparent or opaque voxels that cause this darkening by absorbing the ambient light and not producing any emission [5].

One of the variations of volumetric AO, Local Ambient Occlusion (LAO), is a shading technique that could be used in volume rendering to provide better understanding of volumetric structures by darkening regions less likely to be exposed to ambient light [6]. However, the computation of a darkening factor is expensive because it requires sampling the neighborhood of a voxel and is usually computed prior to visualization. When using interactive volumetric clipping techniques, the ambient occlusion term needs to be recomputed every time the clipping region is modified because any change to the voxel opacity will affect how much neighboring voxels are obstructed. This computation is difficult to achieve in real-time, as can be seen on Fig. 2. In this paper, we propose a new technique called Contextual Ambient Occlusion (CAO) that enables fast recomputation of regions where ambient occlusion is likely to have changed every frame, while maintaining temporal stability.

The main idea of the algorithm is that in LAO, the ambient factor of a voxel is affected only by a small number of surrounding voxels in a certain spherical radius. Therefore, when parts of the volume are clipped, only a small number of voxels have their value affected

---

*e-mail: andrey.titov.1@ens.etsmtl.ca

†e-mail: marta@ap-lab.ca

‡e-mail: simon.drouin@etsmtl.ca

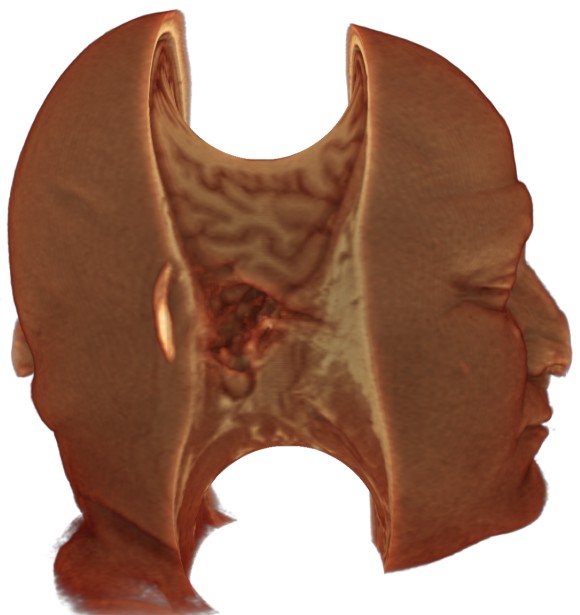

Figure 1: CAO with a concave clipping shape (torus).

by the clipping. In our algorithm, we propose a pipeline that allows us to identify those voxels and perform the LAO recalculation only for them. Fig. 1 illustrates an example of an image that we were able to obtain with this algorithm.

The algorithm was specifically designed for volume ray casting where a perspective transformation (which includes all linear transformations such as translation, rotation, and scaling) could be applied to the clipping region every frame. Therefore, it is particularly well suited for interactive visualization of volumetric data where the user controls the position and rotation of the clipping regions, such as with the interactive clipping technique presented by Joshi et al. [7]. In addition to the fast calculation time, the algorithm achieves high AO quality because it uses the discrete volume rendering integral for opacity calculations, similar to Hernell et al. [6]. However, unlike the implementation described by Hernell et al. [6] where only one ray is cast per frame, the algorithm calculates the final AO factor for all rays every frame, which avoids having a blurry effect that comes from the combination of multiple frames. This makes it possible to use our algorithm even with a significant change of the position, rotation or scale of the clipping region every frame.

In addition to the previous advantages, the new method is compatible with virtual reality (VR) and augmented reality (AR) displays because all the ambient occlusion factor calculations are done in data space. This avoids the problem of screen-space stereoscopic inconsistency [13] and makes the algorithm scale well with an increasing resolution and number of viewports. We evaluated the performance of the algorithm in comparison with a naive implementation where the ambient factor is recalculated for all voxels every frame and

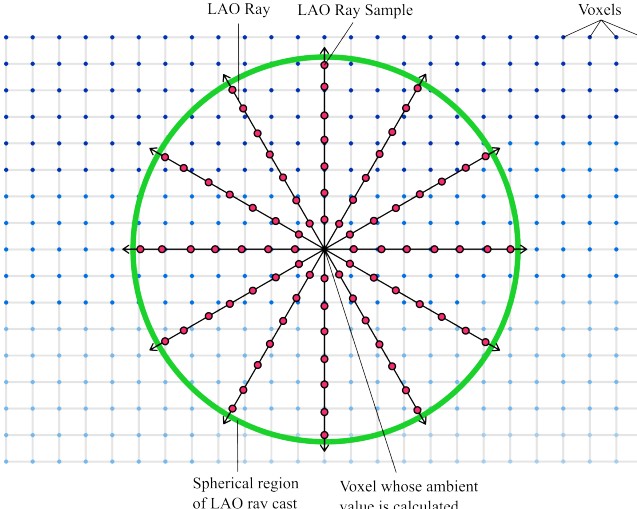

LAO Ray   LAO Ray Sample    Voxels

Spherical region   Voxel whose ambient
of LAO ray cast   value is calculated

Figure 2: Ray cast performed for each voxel to calculate the LAO factor. Each red point represents a tri-linearly interpolated sample taken from the volume. The ambient factors of all rays are then aggregated together to create a single ambient value for the voxel.

found that our algorithm offers a 2-5x speedup. Additionally, if we compare our algorithm to the one proposed by Hernell et al. [6], the latter takes multiple frames (e.g. 6 to 54 to achieve a similar quality to images presented in Fig. 10) to recover from a change of clipping frames, which is unacceptable for interactive VR environments, and our algorithm fixes this problem.

## 2  RELATED WORK

One of the most popular techniques to calculate ambient occlusion that is commonly used in real-time computer graphics is Screen-Space Ambient Occlusion (SSAO), which has the advantages of only requiring a depth map as an input and having good performance due to the fact that all calculations are done in screen-space [1]. However, in volume rendering where a depth map cannot be obtained due to the absence of isosurfaces in the dataset that needs to be visualized, alternate algorithms to estimate the ambient factor have to be used.

One of the first papers to present a real-time implementation of AO-alike volume shading technique was Vicinity Shading [14]. This algorithm first calculates the normal for each voxel, and then sends 3D Bresenham rays in a half-spherical neighborhood at the same side as the normal vector. Each ray traverses the volume until it reaches a voxel whose opacity is higher, and the results of all rays are aggregated together using averaging. This averaged value then represents that vicinity value of the voxel, which can also be interpreted as the ambient factor of the voxel.

Díaz et al. [2] presented an algorithm that speeds up the precalculation part compared to vicinity shading and doesn't require calculating the normal of every voxel. The algorithm uses the concept of 3D summed area tables (SAT) which are used to quickly aggregate and calculate the average opacity of multiple voxels. In the precalculation step, a 3D SAT volume of opacities is constructed from the initial volume and used during rendering to quickly evaluate the average vicinity value for a single voxel with very few texel fetches. The biggest disadvantage of this algorithm is that 3D SATs give only a rough approximation of the ambient value because all voxels used to calculate the vicinity value have equal weight, no matter their distance from the voxel. To improve upon this problem, another algorithm based on the idea of using 3D SATs was presented by Schlegel et al. [12]. This improved algorithm changes how the 3D SAT is used at runtime. Instead of simply estimating the average

Table 1: Comparison of existing volumetric ambient occlusion methods. Quality represents the relative quality of the obtained ambient value. The "Real-time modification to" column indicates what type of modification could be done in real-time within the algorithm without a loss in performance. The red, yellow and green circles represent low, medium and high quality of the ambient factor correspondingly.

| Author(s) | Name | Quality | Real-time modification to: | | |
| | | | Radius | TF | Clipping |
| --- | --- | --- | --- | --- | --- |
| Stewart [14] | Vicinity shading | 🟡🟡⚪ | ✗ | ✗ | ✗ |
| Ropinski et al. [11] | Clustered histograms | 🟡🟡⚪ | ✗ | ✓ | ✗ |
| Díaz et al. [2] | SAT of opacities | 🔴⚪⚪ | ✓ | ✗ | ✗ |
| Hernell et al. [2] | Local Ambient Occlusion | 🟢🟢🟢 | ✓ | ✗ | ✗ |
| Schlegel et al. [12] | Shelled SAT of opacities | 🟡🟡⚪ | ✓ | ✗ | ✗ |
| Engel & Ropinski [3] | Deep-learned AO | 🟡🟡⚪ | ✗ | ✓ | ✓ |
| Ours | Contextual Ambient Occlusion | 🟢🟢🟢 | ✗ | ✗ | ✓ |

vicinity value around the voxel, the algorithm calculates "shells" around the volume where each one has a different distance from the voxel. Then, the values of all shells are aggregated together using the absorption formula presented by Max [8], similar to the standard volume rendering integral. However, even in that case, due to the aggregation of many voxels together, only a rough estimation of the ambient factor is obtained.

Unlike 3D SAT table algorithms which focus on fast precalculation and rendering time, some other algorithms focus on having a high quality of the ambient value by focusing on the physical accuracy of the ambient value calculation. Such is the case of the Local Ambient Occlusion (LAO) algorithm presented by Hernell et al. [6]. This algorithm presents an expensive precalculation step where for each voxel its ambient value is calculated. This calculation consists of casting multiple rays in a spherical pattern around each voxel. Each of the rays samples the opacities in the volume and aggregates them using the absorption formula described by Max [8]. The values of all rays are then averaged together to create a single ambient value for the voxel. The algorithm produces high quality results as further voxels affect the ambient value exponentially less than closer ones, which is more physically accurate.

Some ambient occlusion algorithms focus on being able to properly estimate the ambient factor if the opacity transfer function of the volume changes in real-time. Such is the case of the clustered histograms algorithm presented by Ropinski et al. [11], whose main idea is that voxels with approximately the same neighborhood will have roughly the same ambient factor independently of the transfer function used. In this algorithm, for each voxel, a local histogram of neighboring voxels is created, and then similar histograms are clustered together. This histogram stores that degree of influence that each neighboring voxel has on the main voxel using inverse distance weighting. Consequently, a new 3D texture is filled which contains the cluster index for every voxel. With this information, the ambient factor is calculated in real-time by applying the transfer function on the histogram. In 2021, a different approach was proposed to

estimate the ambient factor with dynamic transfer function change by Engel and Ropinski [3]. In this algorithm, deep learning was used to predict the impact of a change of the transfer function on the ambient value. Many combinations of MRI volumes and transfer functions were created and rendered with Monte Carlo ray tracing, and the resulting voxel ambient values were learned. The created model could then be used to predict the ambient value for any MRI image and any opacity transfer function.

In Table 1, we presented a summary of volumetric ambient occlusion algorithms where we compared them in terms of their ambient factor quality, and we also evaluated what types of real-time changes can be done to the algorithm. In order to rate the quality, we separated all the algorithms into 3 logical groups, ranging from low to high that measure how physically accurate the ambient factor is. The high-quality group encompasses all algorithms that follow the absorption model presented by Max [8] and perform spherical ray casting. The medium-quality group consists of algorithms that provide some variation of inverse weighting for voxels based on their distance from the voxel for which the ambient value is calculated. The low-quality group consists of algorithms that calculate an unweighted average of opacities of neighboring voxels for a single voxel.

## 3 METHODS

The CAO algorithm is able to recalculate in real-time the changes to occlusion that occur in the volume when parts of this volume are clipped, while achieving exactly the same quality as LAO. Because the algorithm simply prevents unaffected voxels from being recalculated, the calculation is always exact and remains stable over successive frames. All of the ambient occlusion computations are done in data space to avoid the stereoscopic screen-space inconsistency problem described by Shi et al. [13], which makes it suitable for VR and AR environments.

### 3.1 Pipeline

The central idea of the CAO algorithm is that we render an inflated version of the clipping shape in two depth buffers, and then we use these buffers to determine which voxels need recomputation of the ambient factor.

The algorithm requires as input a volume $V$ stored in a 3D texture, the opacity transfer function $TF_o$ as well as the clipping mesh $M$. The mesh $M$ should satisfy the requirements described by Weiskopf et al. [16], meaning it has to be either convex, or concave with the limitation that it could be fully rendered with exactly two opacity peeling steps [4]. For example, as indicated in Fig. 1, a concave shape such as a torus can be used in this algorithm because it satisfies this constraint.

We propose a novel approach to obtain the correct ambient factor for all voxels every frame by only recalculating the ambient factor for voxels where it was affected by clipping. We use shadow test volume clipping (see Sect. 3.2) to perform the clipping and dilation maps (see Sect. 3.3) to determine which voxels were affected by clipping.

A visual representation of the pipeline is given in Fig. 3, where the steps executed using a compute shader are marked as C.S., while the steps executed using the rendering pipeline are indicated with R.P.

Steps done before ray casting:

1. (C.S.) The ambient factor is computed for all voxels using LAO spherical raycasting assuming that no clipping is applied on the volume. The resulting AO values are stored in a volume called $V_{PAO}$ and used later.

2. (R.P.) The front and back faces of the clipping mesh are rendered using two depth-only rendering passes and the resulting depth maps are stored in two textures.

3. (C.S.) Dilation maps are computed from the depth maps created in step 1 and stored in two textures. The dilation maps are computed by calculating the Minkowski sum between the depth map and a sphere that represents the LAO rays cast region.

Steps done during rendering:

4. (C.S.) Shadow test volume clipping (see Sect. 3.2) is done for each voxel of the initial volume $V$, and the result is stored in a volume $V_o$ containing the opacity of each voxel after clipping is applied. Additionally, a mask volume $V_m$ is populated that indicates for each voxel if it was clipped, and in the case that it was not, whether its ambient value was affected by clipping.

5. (C.S.) A volume containing the final LAO coefficients $V_{AO}$ for the image is outputted. The volume $V_m$ is read to determine whether the LAO value should be recalculated with a spherical ray cast, or if the initial ambient value stored in $V_{PAO}$ could be used as-is.

6. (R.P.) The initial volume $V$ is rendered using shadow test volume clipping (see Sect. 3.2), and at every ray step the $V_{AO}$ volume is sampled to determine the ambient factor for the current ray sample.

### 3.2 Shadow test volume clipping

In our implementation, we use a clipping algorithm similar to shadow test clipping presented by Weiskopf et al. [16], which we adapted to be used with ray casting volume rendering. It is used during the computation of LAO and to render the clipped volume. Shadow test clipping is a modification of convex volume cutting based on depth clipping described by Weiskopf et al. [16]. The advantage of this algorithm is that it makes it very simple to test for any point inside the volume if it is clipped, by only requiring to sample two depth maps.

In standard depth clipping [16], a convex mesh is inserted in the volume, and the boundary of the mesh is used to separate the clipped parts of the visible ones. In our case, we consider the region outside the mesh as being visible, while the inside part is considered to be invisible, and therefore clipped. This corresponds to depth clipping described by Weiskopf et al. [15], which can be seen as a subtraction of the clipping mesh from the original volume. However, other types of logical operations can be supported too. The algorithm could be adapted to work with volume probing [15] where the region inside the mesh is visible and everything else is invisible. This would correspond to the intersection logical operator between the mesh and the volume. The modification would require performing a contraction instead of a dilation of the clipping mesh in step 3 of Sect. 3.4, as well as inverting the conditions described in Sect. 3.5 in steps 4c and 4e.

In the original implementation of depth-based volume cutting [16], the front and back faces of the clipping mesh are rendered from the point of view of the observer in two different passes, and the created depth buffers $Z_{front}$ and $Z_{back}$ are stored. Then, during rendering, each fragment[1] $f$ is considered as visible only if it passed the following clipping test (assuming that smaller depth values correspond to closer objects to the camera):

$$visible(f) = (z_f \leq z_{front}) \vee (z_f \geq z_{back}), \quad (1)$$

where $visible(f)$ is a Boolean function that indicates if the fragment $f$ is visible (i.e. not clipped), $z_f$ corresponds to the depth

---

[1]Note that the word "fragment" is used to describe samples of the volume because the original clipping algorithms were implemented with texture mapping [16]. In ray casting, "sample" should be used instead to denote information retrieval from a 3D texture.

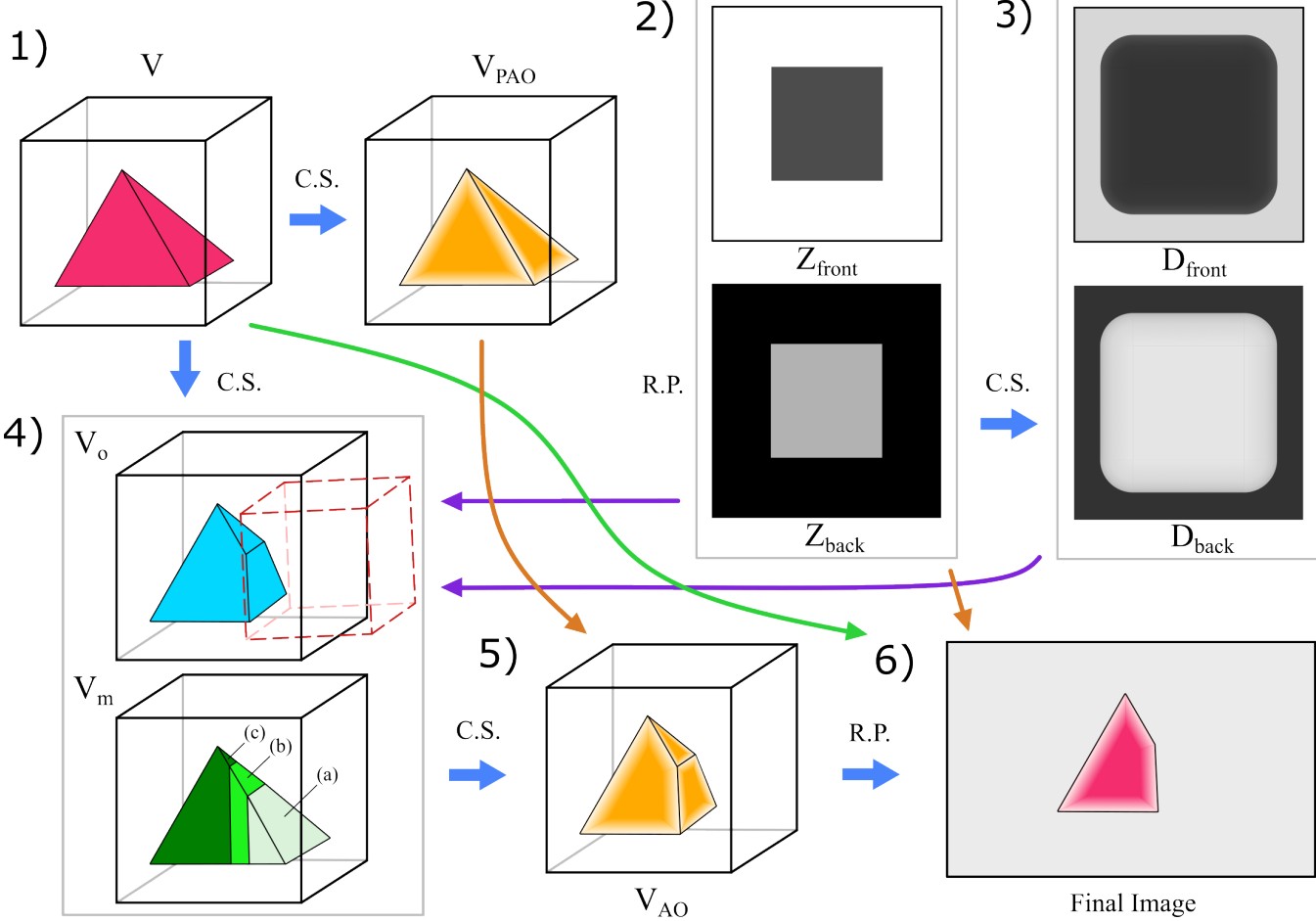

Figure 3: Visual representations of the 6 steps necessary to obtain the volume shaded using CAO. Steps 1-3 are done once before the rendering, while steps 4-6 are done every frame. C.S. represents compute shaders while R.P. stands for the rendering pipeline.

value of $f$, and $z_{front}$ and $z_{back}$ represent the depths within the depth buffers $Z_{front}$ and $Z_{back}$ for the current pixel location of $f$.

Shadow test volume clipping is a modification of the depth-based clipping where the location from which $Z_{front}$ and $Z_{back}$ are computed differs from the location of the observer [16]. In this scenario, these two depth buffers are created from an arbitrary position in the scene with projection parameters that may differ from the projection camera, and we will further refer to it as the clipping camera. Because of this change, when the clipping test is performed on each fragment $f$ during rendering, this fragment should be transformed to the coordinate space of the clipping camera. This technique is referred to as shadow test clipping because the idea of this technique is very similar to shadow mapping [17], where a light source creates a depth map of the scene, and each sample in the scene has to be transformed into the coordinate space of the light source projection camera, and its depth is compared to the one in the shadow map to determine whether the sample is lit or in a shadow. An illustration of shadow test clipping is given in Fig. 4.

In our implementation, we use shadow test volume clipping both during the computation of the volume containing LAO information, but also during the rendering of the volume. As indicated in Fig. 4, we use an orthogonal projection for the clipping camera to obtain the highest precision in the depth buffers. During the rendering, the position and rotation of the virtual camera is synchronized with the clipping mesh, so that the depth maps never change from the point of view of the camera. This makes it possible to precalculate these

depth maps before the rendering and never change them after, unless the geometry of the clipping mesh would dynamically change.

### 3.3 Local Update of LAO

The key idea of CAO is that clipping modifications only affect the ambient factor of a small number of voxels located close to the clipping boundary, so LAO could be recalculated only for the voxels that were affected by clipping. This is illustrated in Fig. 5, where all voxels fall into 3 categories: (a) voxels that were clipped, (b) non-clipped voxels whose ambient factor was affected by the clipping and (c) non-clipped voxels that weren't affected by the clipping. By only recalculating the ambient factor of voxels that fall into category (b), it is possible to significantly speed up the calculation time of LAO without compromising the quality of the final rendered image. In order to do so, the CAO algorithm precalculates the LAO factor for all voxels of a volume before any clipping is applied, and intelligently recalculates LAO only for those affected voxels.

To distinguish which voxels have an ambient factor affected by clipping, a Minkowski sum is calculated between the clipping mesh and a sphere representing the neighborhood considered for LAO calculation, as illustrated in Fig. 6. The radius of the sphere must correspond to the length of the rays sent during the ray casting for each voxel. The mesh created from this Minkowski sum could be used with depth-based mesh clipping to delimit the region of voxels for which LAO should be recalculated. We will further refer to it as the dilation shape.

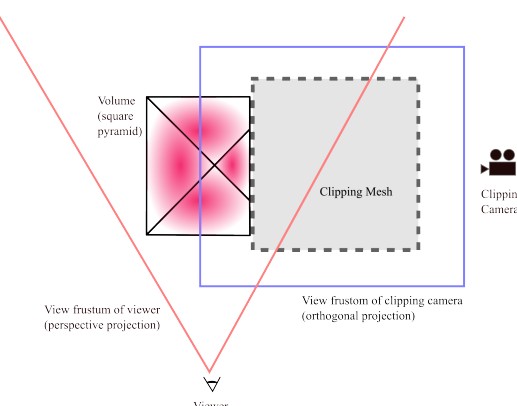

Figure 4: Shadow test volume clipping. The clipping camera is attached to the clipping mesh and renders its front and back faces once to depth buffers. These depth buffers are then used every frame to determine which fragments are clipped by performing a clipping test. Each fragment needs to be transformed to the coordinate space of the clipping camera, sample the depth buffers and then perform the clipping test to determine whether the fragment is visible.

In the actual implementation, this Minskowski sum doesn't need to be explicitly performed and the dilation shape doesn't need to be created in the form of a mesh because doing so is computationally expensive, and a significantly simpler approach could be taken instead. Since the initial clipping mesh already creates depth maps used for depth-based volume clipping, the depth maps of the dilation shape could be computed using them. Therefore, instead of computing a Minkowski sum in 3D, it can be calculated in 2D by adding a half-spherical depth map to the depth maps created by the initial mesh. This sum will create the depth map of the dilation shape which is then used to classify voxels. Since the rendering of the volume will be done with shadow test volume clipping, this operation would only need to be performed once before the rendering begins.

### 3.4 Precomputation of the Dilation Shape Depth Maps

Before starting rendering, three precalculation steps should be completed.

1. **The ambient factor is computed for all voxels in the volume in a spherical neighborhood using LAO and the result is stored in a volume called $V_{PAO}$.** In order to compute the ambient factor for a single ray, the absorption model is used [6]. Thus, for a ray $k$ sent from the voxel $x$ the ambient factor is expressed as:

$$A_k(x) = \frac{1}{M} \sum_{m=1}^{M} \prod_{i=1}^{m-1} (1 - a_i), \quad (2)$$

where $M$ represents the number of samples taken by the ray, $m$ and $i$ are used to iterate through the samples (assuming that smaller numbers corresponds to closer samples), and $a_i$ represents the opacity at the $i$-th sample. To combine the ambient values of all rays, the following formula is used (which performs an averaging):

$$A(x) = \frac{1}{K} \sum_{k=1}^{K} w_k A_k(x), \quad (3)$$

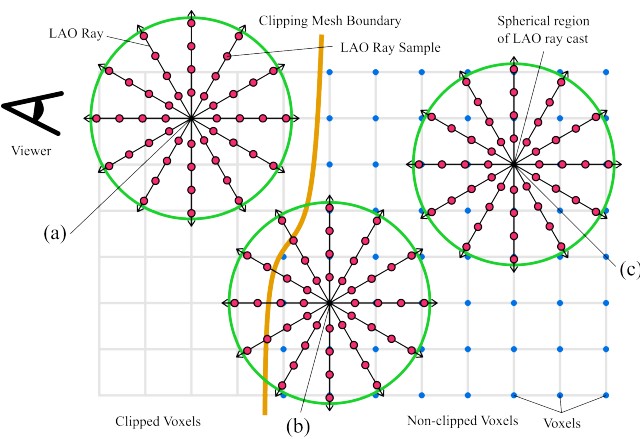

Figure 5: 3 possible categories for the voxel. The green circles represent the spherical neighborhood used to perform the LAO calculation. The orange line represents the boundary of the clipping mesh.

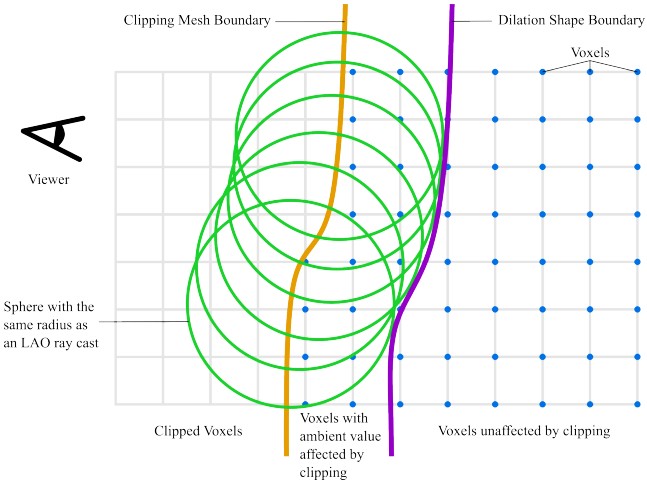

Figure 6: 3 possible categories for the voxel. The green circles represent the spherical neighborhood used to perform the LAO calculation. The orange and the purple lines represent the boundaries of the clipping mesh and the dilation shape correspondingly.

where $K$ represents the number of rays, $k$ is used to iterate through the rays, $A_k(x)$ represents the ambient value of the ray $k$, and $w_k$ is an optional parameter that could be used to perform directional weighting. The obtained ambient factor $A(x)$ ranges from 0 (the voxel is fully occluded) to 1 (the voxel is completely unobstructed).

2. **A "clipping" camera is introduced in the scene which renders the clipping mesh with two depth-only passes to two framebuffer objects (FBOs) using orthogonal projection.** The clipping camera is used to render this mesh from some point in the scene, similar to how a light source generates a depth map in shadow mapping (see Fig. 4). The front and the back faces of the mesh are rendered into each FBO correspondingly, and the value of empty pixels of $Z_{front}$ is inverted. This way, for any voxel in the volume, it can easily be determined if it is clipped by transforming it to the coordinate space of the clipping camera. The two FBOs $Z_{front}$ and $Z_{back}$ are kept without change for all future rendering.

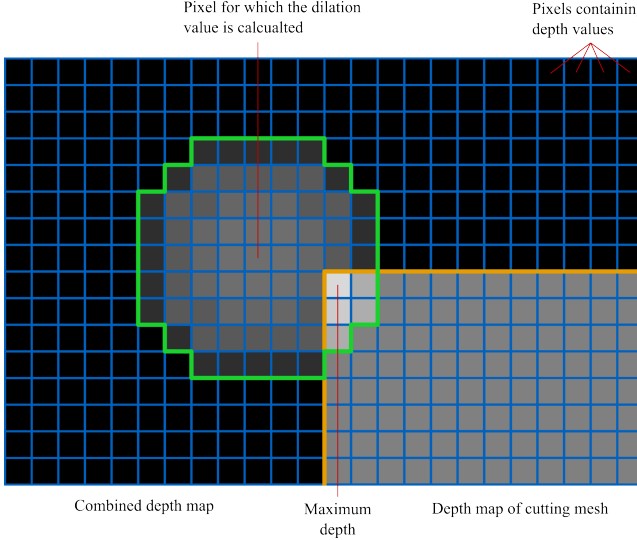

Pixel for which the dilation value is calcualted

Pixels containing depth values

Combined depth map

Maximum depth

Depth map of cutting mesh

Figure 7: Computation performed for every pixel during the Minkowski sum calculation. The algorithm "inflates" the original depth map using a spherical depth map and stores the biggest value that was possible to obtain. Note that here, the creation of $D_{back}$ from $Z_{back}$ is illustrated.

The 3D pose of the clipping mesh relative to the clipping camera should respect some constraints. If the clipping mesh is convex, then it should be positioned in the center of the clipping camera and have a margin around it to leave space for the dilation map generated in step 3. If the mesh is concave, then an additional limitation should be respected, which is that the clipping mesh could be rendered with 2 depth peeling passes from the point of view of the clipping camera. If this limitation is not respected, the clipping mesh would cut regions similar to the convex hull of the concave mesh.

3. **The Minkowski sum of the clipping mesh and the LAO ray cast region is rendered to two additional FBOs.** This Minkoski sum doesn't need to actually be represented with a mesh in order to be projected, it can be calculated in screen space from the two FBOs which contain the original projected mesh. These new textures containing the depth maps of the dilation shape are called $D_{front}$ and $D_{back}$.

To calculate $D_{front}$ from $Z_{front}$, a compute shader needs to be executed that will combine the depth map $Z_{front}$ with a spherical kernel that represents the depth map of a sphere, as illustrated in Fig. 7. The compute shader instantiates a thread for each pixel of the output depth map $D_{front}$, which has the same 2D dimensions as $Z_{front}$. The algorithm requires input the radius of the sphere called $r_{xy}$ which is measured in pixels, as well as the $z$-offset $r_z$ which represents a depth in the range $[0, 1]$. These values are calculated by projecting the dilation radius in the coordinates of the carving camera and measuring the pixel length, as well as the depth. This dilation radius depends on the length of the rays used in step 1. Given that the thread calculated the dilated depth value of pixel $(x, y)$, the following steps are executed for each pixel in the circular neighborhood $r_{xy}$ around this pixel. To index each of those pixels, $(i, j)$ is used:

(a) The depth value within $Z_{front}$ of the pixel located at position $(i, j)$ is fetched and stored in the variable $z_{mesh}$.

(b) The relative depth value $z_{sphere}$ of the dilation sphere is

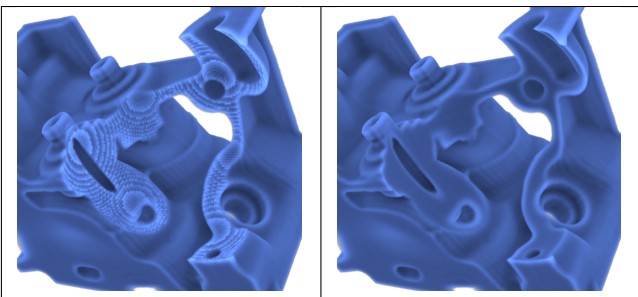

Figure 8: Comparison between CAO images created (left) without anti-aliasing and (right) with anti-aliasing with $N = 32$ . Both images are rendered with a 54-ray spherical raycast.

calculated using the following formula:

$$z_{sphere} = r_z * \sqrt{1 - \left(\frac{norm((x-i, y-j))}{r_{xy}}\right)^2} \quad (4)$$

(c) The depth value $z_{sphere}$ is subtracted from $z_{mesh}$ and the result is stored in a variable $z_{dilated}$.

(d) The smallest depth value $z_{dilated}$ obtained for any $(i, j)$ voxel is then stored in $D_{front}$ for the pixel $(x, y)$.

Then, the same process is repeated for the buffers $Z_{back}$ and $D_{back}$ with the exceptions that in step 3c, $z_{sphere}$ is added to $z_{mesh}$ and the maximum instead of the minimum $z_{dilated}$ value is stored in $D_{back}$.

### 3.5 Steps performed each frame

Each frame, the following computation is performed:

4. **A compute shader clips the initial volume $V$ and outputs a 3D texture $V_o$ where all the clipped voxels are set to be transparent, while the visible voxels hold the opacity of the voxel. Additionally, the mask volume $V_m$ is populated.** Here, the computer shader code is executed once for each voxel, and outputs either 0 (for zero opacity or full transparency), the original opacity of the voxel as if no clipping was applied, or the anti-aliased opacity value. Fig. 8 illustrates the importance of using anti-aliasing when filling the $V_o$ volume.

To calculate the anti-aliased opacity value, multisample anti-aliasing (MSAA) is used. First, for each voxel, one sample is taken in its center, and 8 samples are taken at the corners of the voxel. The algorithm executed for each sample is shadow test clipping which also includes populating the mask volume $V_m$ in addition to filling the opacity volume $V_o$:

(a) The sample is transformed to the coordinate space of the clipping camera. The resulting $xy$ coordinates will correspond to a pixel position in the FBO, while the $z$ component corresponds to the depth of the sample in the clipping camera coordinate space.

(b) The near ($Z_{front}$) and far ($Z_{back}$) depth maps obtained in the precalculation step 2 are sampled at the 2D position obtained in step 4a, which gives two depth values.

(c) The depth of the sample obtained in step 4a is compared to the depths obtained in step 4b. If the depth of the sample is between the depth values of $Z_{front}$ and $Z_{back}$, the sample is considered to be invisible, otherwise it is

considered as visible. If the voxel is visible, a counter $C_{visible}$ is incremented that keeps track of the number of visible samples.

(d) The $D_{front}$ and far $D_{back}$ dilation maps obtained in the precalculation step 3 are sampled at the 2D position obtained in step 4a, which gives two depth values.

(e) The depth of the sample obtained in step 4a is compared to the depths obtained in step 4d. If the depth of the sample is between the depth values of $D_{front}$ and $D_{back}$, a counter $C_{dilated}$ is incremented that keeps track of the number of samples inside the dilation shape.

After this, a check is made to determine if antialiasing should be used:

- If the algorithm determines that the 9 samples are either all visible or all invisible, then it means that the voxel was not affected by anti-aliasing, and its value is used directly. In that case, if all samples are visible, the opacity transfer function is applied to the original voxel value and saved in the $V_o$ volume. If all samples are invisible, then an opacity of 0 is stored in $V_o$ for the voxel.

- If some of the samples are visible and some are not, then $N$ additional samples are taken at predefined positions in the $[-1,-1,-1]$ to the $[1,1,1]$ neighborhood of the voxel, for a total of $N+9$ samples. For these samples, the steps (a), (b) and (c) are repeated again, continuing to increment the counter of the number of visible samples $C_{visible}$. After this calculation is done, the anti-aliased opacity is obtained using the following formula:

$$O_{AA} = \frac{C_{visible}}{N+9} \, O_{voxel} \qquad (5)$$

where $C_{visible}$ represents the number of visible samples, $N$ represents the number of anti-aliasing samples, and $O_{voxel}$ represents the opacity of the voxel after the opacity transfer function was applied to it. $O_{AA}$ is further saved into the $V_o$ volume for the corresponding voxel.

In addition to calculating the anti-aliased opacity for each voxel, the compute shader also outputs a 3D mask $V_m$ that would indicate to which of the following categories the voxel corresponds: clipped (category $a$), not clipped, but LAO factor affected by clipping (category $b$) and not affected by clipping (category $c$). To determine this, a simple check is executed on the counters:

- If $C_{dilated}$ is equal to 0, it means that the voxel is completely outside the dilation shape, which means that it corresponds to category $c$, and a value of 1 is stored in $V_m$ for the current voxel.

- Otherwise, if $C_{visible}$ is bigger than 0, it means that the voxel is outside the clipping shape but inside the dilation shape, so it corresponds to category $b$, and a value of 0.5 is stored.

- Otherwise, the voxel is completely clipped (category $a$), so a value of 0 is stored.

5. **A compute shader reads the mask volume, calculates the LAO factor in a different manner depending on the mask value $V_m$, and outputs the LAO factor to the $V_{AO}$ volume.** If the voxel is of category $a$, then no calculation is performed and the ambient factor that is saved is 1. If the category is $c$,

Table 2: Scenes tested during the evaluation.

| Scene | Resolution | Samples Per Ray | Clipping Mesh | % Voxels Recalcu- lated |
|---|---|---|---|---|
| Engine | $256 \times 256 \times 110$ | 20 | sphere | 14.94 |
| Beetle | $416 \times 416 \times 247$ | 40 | cube | 20.24 |
| Skeleton | $512 \times 512 \times 512$ | 50 | rectang. prism | 16.13 |

then no LAO calculation is performed either, and instead the precalculated $V_{PAO}$ volume is sampled and its value is copied in the output $V_{AO}$ volume as-is. Finally, if the category is $b$, only then the LAO calculation is performed for the voxel.

For each voxel in category $b$, rays are sent in $K$ predefined directions, similarly to how it was made in step 1 for the volume $V_{PAO}$. The ambient factors of all rays are then averaged, and this value is written in the output 3D texture $V_{AO}$.

6. **The volume $V$ is rendered with a ray casting algorithm, and for each sample of each ray, the $V_{AO}$ volume is sampled to determine the ambient factor.** Here, the volume is rendered using standard ray casting, and at every sample of each ray, shadow test volume clipping using the $Z_{front}$ and $Z_{back}$ depth maps is used to determine if the sample is visible or not. If it is visible, then the $V_{AO}$ volume is sampled.

## 4 EVALUATION

To evaluate the CAO algorithm, 3 scenes were created, each featuring a different dataset with different parameters, as indicated in Table 2. The volumes were taken from the ImageVis3D 1 dataset. In each scene, a volume was rendered, from which a region was clipped away. We used a single-color transfer function for each volume to better highlight the effect of LAO rendering on the dataset. The 3 rendered volumes are demonstrated in Fig. 10 with two different LAO ray numbers: low (6 rays) and high (54 rays). A close-up view is given in Fig. 11. Regarding the number samples $N$ used for anti-aliasing, we used 32 for a total of 41 samples including the 9 original ones taken for all voxels. To execute all our tests, we used a Windows 10 machine with an AMD Ryzen 7 5800X CPU, 32 GB of RAM and an AMD RX 6700XT GPU with 12 GB of VRAM.

First, we evaluated the percentage of voxels that would need to be recalculated in our CAO algorithm, which we have also written in Table 2. This was measured by comparing the number of voxels within the $V_m$ volume that were marked as requiring recalculation compared to the total number of voxels in the volume. As can be seen from that table, this percentage is relatively low, hovering around 15-20%. Note that the samples of the LAO rays have a distance of 1 mm between each other, which also corresponds to the Euclidian distance between any two neighboring voxels.

Second, we measured the rendering performance of the CAO algorithm and compared it to a naive implementation where LAO was recalculated for all voxels every frame, as can be seen in Table 3. In the naive implementation, the $V_m$ volume as well as the depth maps of the dilation shape $D_{front}$ and $D_{back}$ were not calculated. However, the output image produced by the naive LAO algorithm is exactly the same as the CAO one. Additionally, we compared the performance of the algorithm with Solid Color rendering where the same color was applied to all samples instead of LAO, but this implementation also featured shadow test volume clipping.

Third, we determined the computation cost breakdown when calculating a single frame of each of the 3 scenes using the LAO and CAO algorithms, with 6 and 54 rays, as can bee seen in Fig. 9. The graphs demonstrate that in terms of the amount of calculations, the

Table 3: Mean frames per second (FPS) for each scene. The speedup represents the ratio of the framerate of our algorithm (CAO) compared to a naive implementation (LAO).

| Scene | Number of Rays | Solid Color | LAO - FPS | CAO - FPS | Speedup of CAO |
|---|---|---|---|---|---|
| Engine | 6 | 272.846 | 111.549 | 184.902 | 1.66 |
| | 14 | | 67.910 | 155.245 | 2.29 |
| | 26 | | 42.831 | 126.378 | 2.95 |
| | 54 | | 22.543 | 85.363 | 3.79 |
| Beetle | 6 | 128.288 | 15.497 | 42.581 | 2.75 |
| | 14 | | 7.509 | 25.581 | 3.41 |
| | 26 | | 4.192 | 16.063 | 3.83 |
| | 54 | | 2.012 | 8.526 | 4.24 |
| Skeleton | 6 | 95.142 | 4.316 | 17.514 | 4.06 |
| | 14 | | 2.020 | 9.594 | 4.75 |
| | 26 | | 1.107 | 5.729 | 5.18 |
| | 54 | | 0.529 | 2.825 | 5.34 |

longest step to perform along those described in Sect. 3.5 is step 5, where the spherical ray cast is performed. Further, with increasing dimensions of the volume that is rendered, the proportion of the step 5 computations becomes even larger, overshadowing steps 4 and 6 in terms of calculation time.

Finally, knowing that the algorithm produces exactly the same image as LAO, to make sure that our implementation is correct, we compared the screenshots produced by the LAO and CAO algorithms for the 3 scenes. We have created screenshots using 6, 14, 26 and 54 ray configurations and we made a per-pixel comparison between the LAO and CAO images for each configuration. We have found that every pair of pixels had the exact same *RGB* components, confirming the correctness of our implementation. This result was expected because our CAO algorithm dynamically determines every frame what voxels could have a LAO value that is affected by clipping, and performs a recalculation on these voxels. Thus, with a correct implementation, it is impossible to obtain a different AO value for LAO and CAO.

## 5 DISCUSSION

The most important advantage of our algorithm is that it offers exactly the same quality of image as the LAO algorithm presented by Hernell et al. [6], while reducing AO factor recomputation time when clipping parameters are modified. This algorithm produces a physically accurate ambient factor, as it follows the absorption model described by Max [8]. We propose a more sophisticated rendering pipeline for the LAO algorithm that is able to take care of carving within the volume while maintaining proper recalculation of the ambient factor. Additionally, our algorithm supports clipping with all convex shapes and concave shapes that can be rendered with two opacity peeling steps [4].

When compared to LAO, it can be seen from Table 3 that our algorithm offers a significant speedup, with 2-5x the frame rate compared to a full recalculation of all voxels. It can also be seen that the speedup is higher for volumes with a higher 3D resolution and with a higher number of rays. This is expected because in those cases the total amount of computation necessary for the spherical ray cast becomes larger, so cutting down the amount of computation at this step yields the best speedup, as can be seen in Fig. 9. For example, for our largest dataset (Skeleton), for the 26 and 54 ray cases, the speedup was 5.18 and 5.34 correspondingly. Thus, our algorithm is especially useful when a high-quality ambient factor needs to be calculated every frame with a volume with high dimensions.

Additionally, since the calculation is done in data space, it avoids the stereoscopic screen-space inconsistency described by Shi et al. [13], which can result in viewer discomfort. This inconsistency

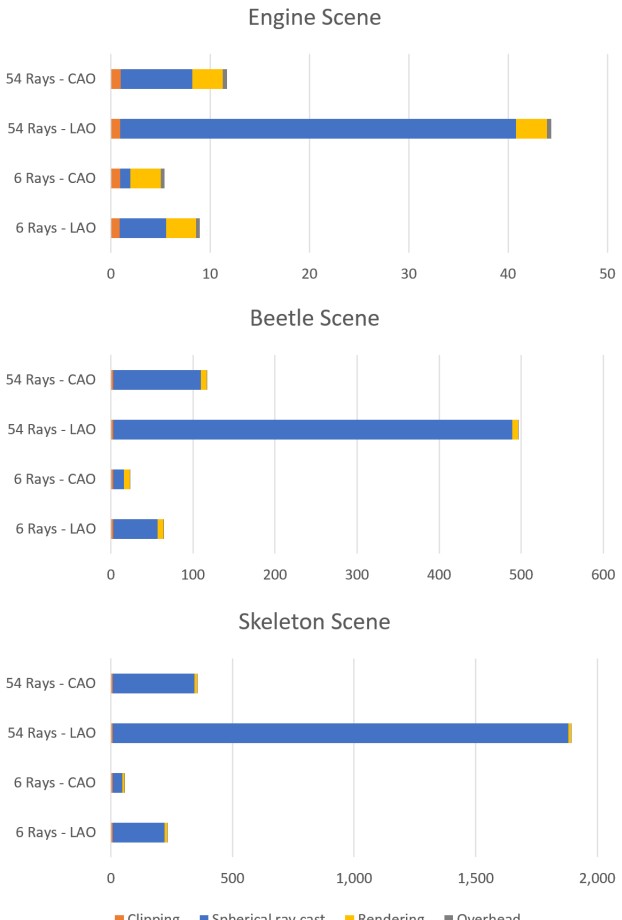

Figure 9: Runtime computation cost breakdown for a single frame using the LAO and CAO algorithms, in milliseconds (ms). "Clipping" corresponds to the step 4 of the algorithm explained in Sect. 3.5, "Spherical ray cast" corresponds to step 5, "Rendering" corresponds to step 6, and "Overhead" represents time spent by the Unity engine on calculations not related to AO.

exists in screen-space algorithms such as SSAO due to the fact the left and right eye view slightly different parts of the dataset, which might consequently lead to a different calculated ambient value for the same object in each eye.

The algorithm that we presented is particularly suitable for interactive environments where the position and rotation of the clipping region might change every frame. Unlike the algorithm presented by Hernell et al. [6] in which only one LAO ray is cast per frame, in our implementation a full ray cast is performed every frame. Because of this, the user always perceives the final LAO image without having blurring or visual artifacts from previous frames, which makes it more suitable for scenarios where the clipping region is controlled by direct user input. Such is the case of interactive VR and AR environments where the clipping region is controlled with hand gestures or controllers with 6 degrees of freedom, so small changes to the clipping region could occur every frame due to hand shaking. In addition to this, the AO calculation performance scales well with increased number of viewports and the resolution in each of the viewports. This is due to the fact that CAO is calculated in data space, but also because the rendering of the volume is done with shadow test volume clipping, which avoids needing to generate the depth

map for every viewport like it is done by Weiskopf et al. [16]. This further makes the algorithm suitable for VR and AR displays which may have more than one viewport and a relatively high resolution for each eye.

The main drawback of the algorithm is that it requires a significant amount of video memory, especially when rendering volumes with a high spatial resolution. As can be seen in Fig. 3, there are 5 volumes with the same dimensions as the dataset that have to be kept in video memory simultaneously. Another drawback is that any change to the opacity transfer function would require a recomputation of the volume $V_{PAO}$ that stores the precalculated LAO values. In that case, the framerate of the algorithm would decrease to the one shown for the LAO algorithm in Table 3. However, as can be seen from this table, even in that case the framerate will remain highly interactive, and only the configurations where the dataset has high dimensions and a high number of rays would be severely affected.

## 6 CONCLUSION

In this paper we have presented a novel volumetric ambient occlusion algorithm that was optimized to work with real-time clipping of the volume. In this algorithm, we determine during each frame which voxels' ambient value was affected by clipping and we perform an LAO recomputation only on these voxels. Since those affected voxels are located in a small radius around the clipping shape, their number is significantly lower than the total number of voxels in the volume. Our algorithm is able to find these voxels by calculating the dilated depth map of the clipping volume, and then by using shadow test volume clipping to determine if the voxels require recomputation. The algorithm offers a 2-5x speedup over recalculating the ambient factor of all voxels every frame. Additionally, the algorithm is suitable for interactive environments since the final image is generated every frame, and it is suitable for VR and AR environments since it doesn't have any stereoscopic inconsistency.

In the future, more tests should be made to quantify the effect of different parameters on the quality of the rendering. The effect of the number of rays used during spherical ray casting should be evaluated to determine the right compromise between the quality of the rendered image and the performance. Additionally, the effect of the resolution of the FBOs of the clipping camera needs to be evaluated both in terms of the quality of the created image, as well as the performance. Moreover, the use of more complex concave meshes could be explored. With these meshes, $P$ depth peeling passes may be required, in which case $P$ depth and dilation buffers should be used, similar to how concave mesh clipping was described by Weiskopf et al. [16]. Further, a more rigorous evaluation should be done to determine how our algorithm combined with interactive clipping helps the perception of volumetric data compared to more basic volume rendering techniques such as Phong shading [10] or using a simple 1-dimensional transfer function.

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

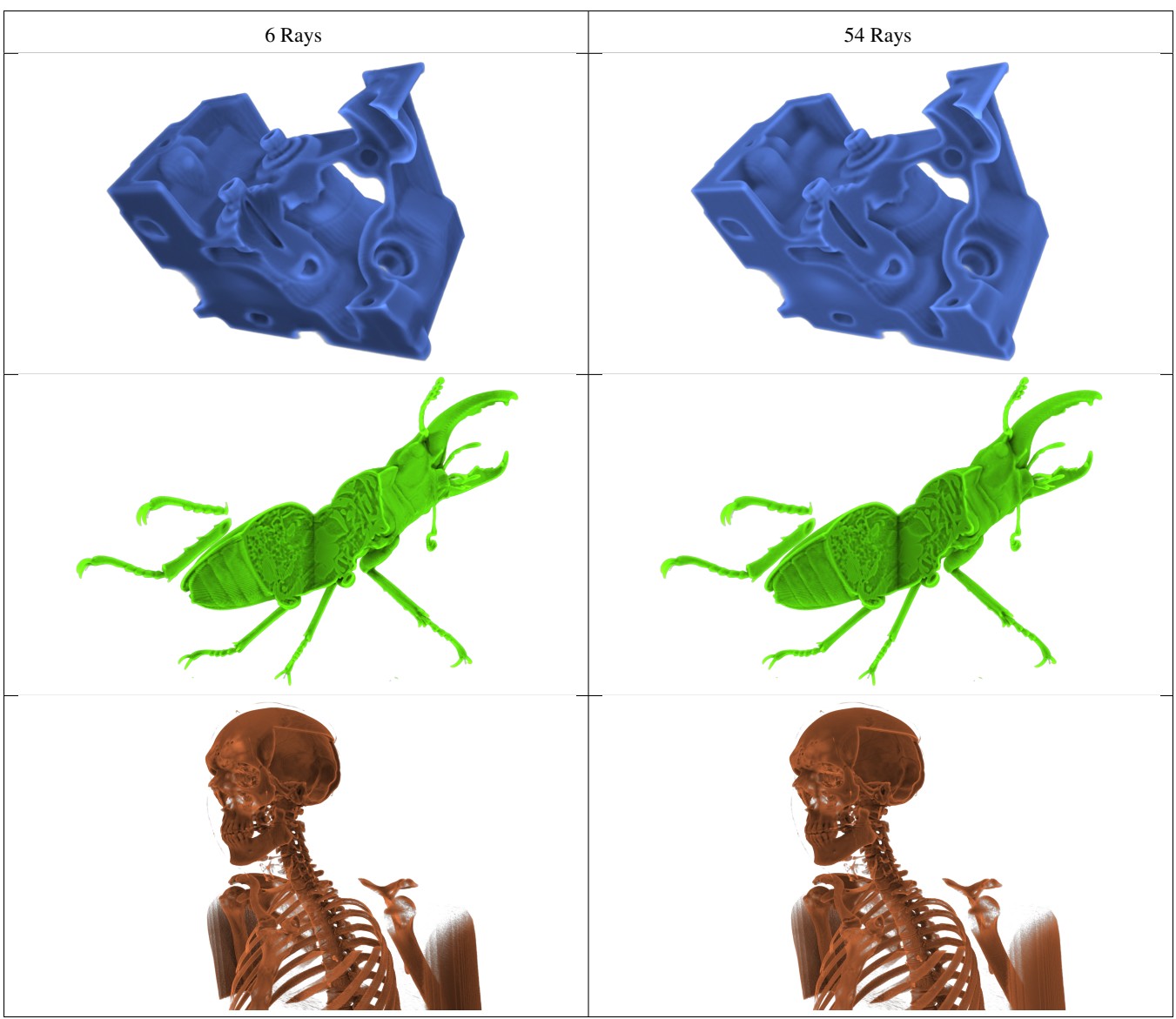

Figure 10: Comparison between the images created using CAO for the 3 scenes using a 6-ray and a 54-ray spherical ray cast. The top, middle and bottom rows demonstrate the Engine, Beetle and Skeleton scenes correspondingly.

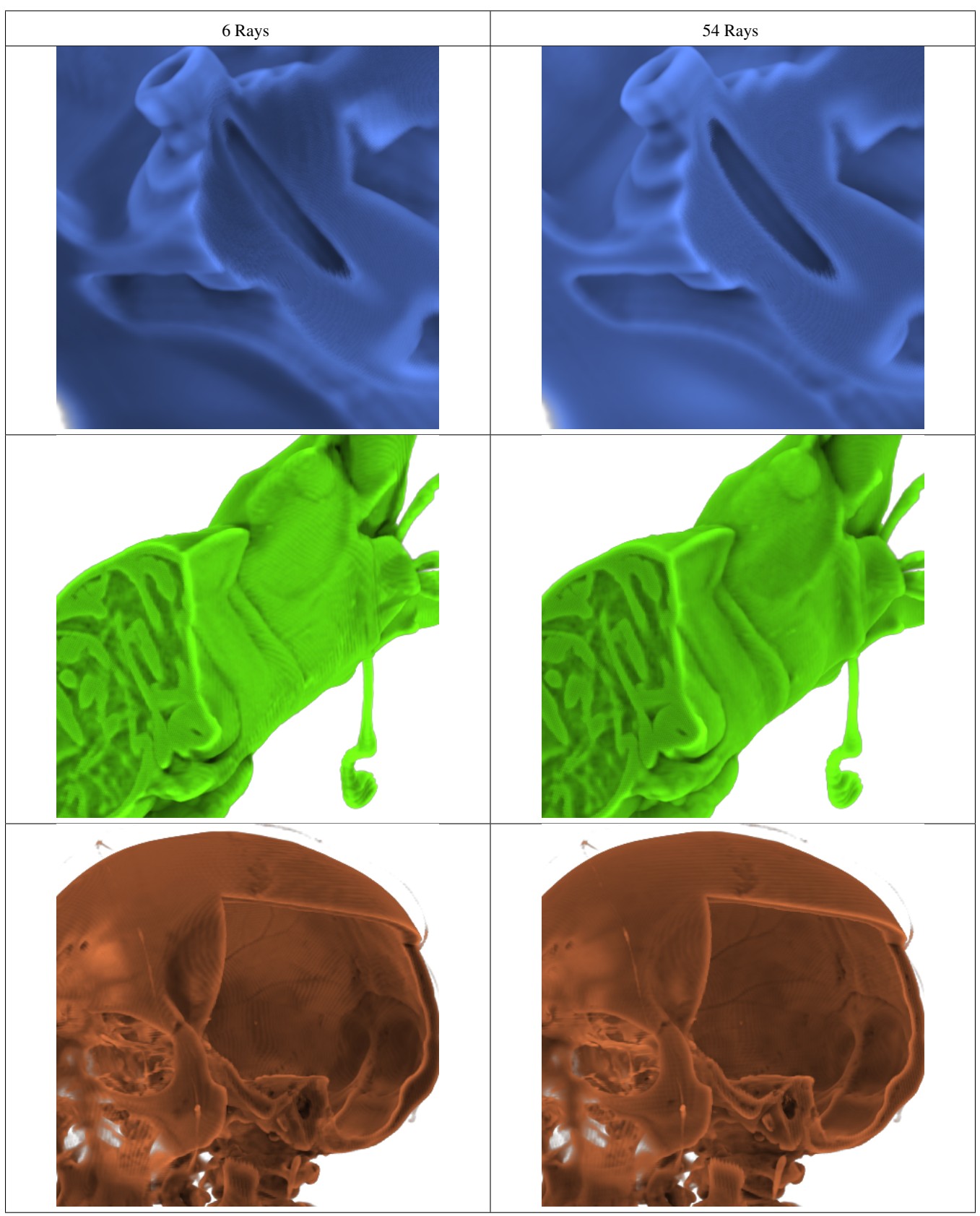

Figure 11: Close-up of the images created using CAO for the 3 scenes using a 6-ray and a 54-ray spherical ray cast. The top, middle and bottom rows demonstrate the Engine, Beetle and Skeleton scenes correspondingly.