# OpenReview forum: "Contextual Ambient Occlusion"
_graphicsinterface.org/Graphics_Interface/2023/Conference — GI 2023_

### Official Review · Reviewer_nPVt · 2022-12-28
**This paper proposed a volumetric AO method for interactive clipping. The strength is the performance gain obtained by partial update, the weakness is evaluation.**

**Rating:** 6
**Confidence:** 4

**Review:**

This paper proposed a volumetric AO method for interactive clipping. The main idea is to exploit the fact that clipping only affects part of the voxels to reduce the required recomputation of LAO values. The novelty of the proposed method is marginal, as it is largely an extension of LAO. The 1.7-8.5 times performance gain is a decent one. The strength of the paper is the performance gain obtained by the partial update.  The weakness is evaluation. Side-by-side comparison is neccessary to justify the claim of "similar quality to Local Ambient Occlusion" and a breakdown of runtime computation cost is good for the readers to better understand the performance gain.

In all, I am slightly positive in accepting this paper for GI 2023.

---

### Official Review · Reviewer_f8BB · 2023-01-09
**A paper with limited technical contributions but a large rendering speed-up**

**Rating:** 6
**Confidence:** 4

**Review:**

This paper presents an optimization of ambient occlusion for direct volume rendering, which computes local ambient occlusion (LAO) with a reduced computational burden (e.g., 1.6x to 8.4x speed up over the original algorithm).

The presented optimization relies on the precomputation of an existing algorithm (LAO) at each voxel and then recomputes the ambient occlusion values only in the voxels where the values can be changed. While the precomputation (and saving those values to the memory) increases the memory overhead, its computational gain is meaningful (e.g., usually more than 2x over the LAO). Like a typical precomputation-based method, it should recompute all voxels using the original algorithm (LAO) when the transfer function changes. This limitation is well-discussed.

On the negative side, the technical contribution of the paper is incremental. It discusses an efficient implementation of a relatively old method (LAO), and I am not sure whether or not accelerating the LAO, which is a crude approximation of global illumination, is one of the main research problems for direct volume rendering.

As a suggestion, I would like to see a discussion on the temporal stability of the presented idea. As the paper argues for the usage in AR and VR, temporally stable results are necessary.

As a minor note, I need clarification about whether the presented optimization is an approximation of the original LAO or not. For example, in the abstract, it is written that the method produces a similar result to LAO, but in the discussion, it is written that the result from this method and the original one is the same. Which one is correct?

Overall, the paper proposes an efficient implementation (recomputation-based optimization) of a previously-proposed ambient-occlusion scheme for DVR and demonstrates a practical gain. While this reviewer is not fully convinced in terms of the novelty and importance of the problem, I like the practical advancement.

---

### Official Review · Reviewer_fSHs · 2023-01-13
**A simple and effective approach for accelerating ambient occlusion computation**

**Rating:** 7
**Confidence:** 2

**Review:**

This paper presents a novel technique to compute local ambient occlusion that supports real-time geometry clipping. The main idea is to leverage the fact that LAO value will only change if nearby geometry gets clipped. Thus, one only requires local updates to the LAO value, instead of a global recomputation. This observation leads to 2-5x speedups.

I am leaning towards accepting this submission. The idea is simple and effective. The paper is well-written and well-evaluated. I especially appreciate a detailed discussion on the related work and those didactic figures for clarifying details of the proposed approach. I believe such an incremental updates on AO could be important for interactive applications, or even for streaming applications.

Perhaps I missed some important points here. But I wonder why the clip mesh needs to be represented as two depth map (front and the back)? Could one use a 3D mesh or an implicit shape to represent the clip mesh, and the use the offset surface the represent the influenced region?

Although this paper solely focuses on clipping geometry, I wonder whether the same idea could be used for other boolean operations, such as union or intersection?

Here are some typos:
- Sec 1: enable better
- Sec 1: real time
- Sec 1: rotation and scaling
- Sec 3.2 during computation
- Sec.6: buffers[

---

### Meta-Review · Area_Chair_jGwC · 2023-01-16

**Recommendation:** 6
**Confidence:** 4

**Metareview:**

All the reviewers agree that the proposed idea can be a simple and practical solution for accelerating ambient occlusion computation. The reviewers especially liked the performance improvements (more than 2X over the baseline) by the presented partial update of the AO.

We encourage the authors to incorporate the following changes into the final paper.
1) Motivate the use of the two depth maps,
2) Discuss the temporal stability of the proposed method,
3) Show if the results of the proposed method are similar to those of the LAO using a side-by-side comparison,
4) Show a timing breakdown of the new technique.

Other minor comments in the reviews need to be addressed for the final version.